# Cohabiting family members share microbiota with one another and with their dogs

Se Jin Song[1], Christian Lauber[2], Elizabeth K Costello[3], Catherine A Lozupone[4†b], Gregory Humphrey[2], Donna Berg-Lyons[2], J Gregory Caporaso[5,6], Dan Knights[7,8], Jose C Clemente[4†a], Sara Nakielny[9], Jeffrey I Gordon[10], Noah Fierer[1,2], Rob Knight[11,12]*

[1]Department of Ecology and Evolutionary Biology, University of Colorado, Boulder, Boulder, United States; [2]Cooperative Institute for Research in Environmental Sciences, University of Colorado, Boulder, Boulder, United States; [3]Department of Microbiology and Immunology, Stanford University School of Medicine, Stanford, United States; [4]Department of Chemistry and Biochemistry, University of Colorado, Boulder, Boulder, United States; [5]Department of Computer Science, Northern Arizona University, Flagstaff, United States; [6]Argonne National Laboratory, Institute for Genomics and Systems Biology, Argonne, United States; [7]Department of Computer Science and Engineering, University of Minnesota, Minneapolis, United States; [8]BioTechnology Institute, University of Minnesota, Saint Paul, United States; [9]Department of Biochemistry and Biophysics, Howard Hughes Medical Institute, University of California, San Francisco, San Francisco, United States; [10]Center for Genome Sciences and Systems Biology, Washington University School of Medicine, St. Louis, United States; [11]Department of Chemistry and Biochemistry, Howard Hughes Medical Institute, University of Colorado, Boulder, Boulder, United States; [12]Biofrontiers Institute, University of Colorado, Boulder, Boulder, United States

*For correspondence:
rob.knight@colorado.edu

†Present address: [a]Mount Sinai School of Medicine, New York, United States; [b]Division of Biomedical Informatics and Personalized Medicine, Department of Medicine, Anschutz Medical Campus, University of Colorado, Denver, United States

Competing interests: The authors declare that no competing interests exist.

**Abstract** Human-associated microbial communities vary across individuals: possible contributing factors include (genetic) relatedness, diet, and age. However, our surroundings, including individuals with whom we interact, also likely shape our microbial communities. To quantify this microbial exchange, we surveyed fecal, oral, and skin microbiota from 60 families (spousal units with children, dogs, both, or neither). Household members, particularly couples, shared more of their microbiota than individuals from different households, with stronger effects of co-habitation on skin than oral or fecal microbiota. Dog ownership significantly increased the shared skin microbiota in cohabiting adults, and dog-owning adults shared more 'skin' microbiota with their own dogs than with other dogs. Although the degree to which these shared microbes have a true niche on the human body, vs transient detection after direct contact, is unknown, these results suggest that direct and frequent contact with our cohabitants may significantly shape the composition of our microbial communities.

## Introduction

Recent studies of the human microbiota have focused on multiple body sites in unrelated adults (*Costello et al., 2009*; *Human Microbiome Project Consortium, 2012*) or on a single body site, such as the gut, in single individuals over time (*Koenig et al., 2011*) or family units, including

**eLife digest** The human body is home to many different microorganisms, with a range of bacteria, fungi and archaea living on the skin, in the intestine and at various other sites in the body. While many of these microorganisms are beneficial to their human hosts, we know very little about most of them. Early research focused primarily on comparing the microorganisms found in healthy individuals with those found in individuals suffering from a particular illness. More recently researchers have become interested in more general issues, such as understanding how these collections of microorganisms, which are also known as the human microbiota or the human microbiome, become established, and exploring the causes of similarities and differences between the microbiota of individuals.

We now know that the communities of microorganisms found in the intestines of genetically related people tend to be more similar than those of people who are not related. Moreover, the communities of microorganisms found in the intestines of non-related adults living in the same household are more similar than those of unrelated adults living in different households. We also know that the range of microorganisms found in the intestine changes dramatically between birth and the age of 3 years. However, these studies have focused on the intestine, and little is known about the effect of relatedness, cohabitation and age on the microbiota at other body sites.

Song et al. compared the microorganisms found on the skin, on the tongue and in the intestines of 159 people—and 36 dogs—in 60 families. They found that co-habitation resulted in the communities of microorganisms being more similar to each other, with those on the skin being the most similar. This was true for all comparisons, including human pairs, dog pairs and human–dog pairs. This suggests that humans probably acquire many of the microorganisms on their skin through direct contact with their surroundings, and that humans tend to share more microbes with individuals, including their pets, with which they are in frequent contact. Song et al. also discovered that, unlike what happens in the intestine, the microbial communities on the skin and tongue of infants and children were relatively similar to those of adults. Overall, these findings suggest that the communities of microorganisms found in the intestine changes with age in a way that differs significantly from those found on the skin and tongue.

those with mono- and dizygotic twin pairs (*Turnbaugh et al., 2008*; *Yatsunenko et al., 2012*). Genetically related individuals, regardless of whether they cohabitate or not at the time of sampling, tend to share more of their gut (fecal) microbes than unrelated individuals (*Caugant et al., 1984*; *Zoetendal et al., 2001*; *Stewart et al., 2005*; *Rajilić-Stojanović et al., 2007*; *Turnbaugh et al., 2008*). However, monozygotic twins are not significantly more similar than dizygotic twins (*Turnbaugh et al., 2008*), indicating that this effect may be influenced by more than genetic similarity. In a study of US teenage mono- and dizygotic twins and their biological parents, we observed that the composition of the fecal microbiota of teens was more similar to that of their parents than unrelated adults, and as similar to that of their fathers as their mothers (*Yatsunenko et al., 2012*). Moreover, mothers and fathers shared more similar bacterial communities in their guts compared to unrelated individuals living in other households (*Yatsunenko et al., 2012*), indicating that a shared environment or lifestyle (e.g., contact with same microbial sources or diet) affects the similarity of the fecal microbiota. Family members may also share intestinal bacteria with their household pets (*Caugant et al., 1984*). Because we leave microbes from our bodies on the surfaces we touch (*Fierer et al., 2010*; *Flores et al., 2011*) and at least a moderate level of microbial exchange is facilitated by direct contact, it is conceivable that our body site-associated microbial communities are shaped in part by our surroundings and those we contact on a daily basis. Whether similar patterns to those mentioned exist within non-gut body sites, whether body sites respond differently to factors such as cohabitation and family structure, and how these patterns change with host age remain unknown.

To test the hypothesis that more microbes are shared between individuals who share a greater number of potential microbial sources, we examined the extent to which microbiota are shared among members of households composed of cohabiting heterosexual adults with and without

children (offspring), and with and without dogs. If our hypothesis were supported, then cohabiting family members would have microbiota more similar to each other than to members of different households. Furthermore, cohabiting couples with either children or dogs would share more microbial taxa in one or more of their body habitats than those without either, because such households contain, in addition to a shared environment, additional shared sources of potentially unique microbes with which couples are in close contact.

## Results and discussion

### Study design

We sampled 159 individuals comprising 17 families with cohabiting children aged 6 months to 18 years, 17 families with one or more dogs but no children, 8 families with both children and dogs, and 18 families with neither children nor dogs. Each family consisted of at least two cohabiting adults (which we define as 'partners' or 'couples') between the ages of 26 and 87 years, and all children included in this study were biologically related to and cohabited with the focal couple. Sampling was performed as described in *Costello et al. (2009)*. For humans, fecal, oral (dorsal tongue), forehead, and right and left palm communities were sampled (n = 5 samples per individual all taken at a single time point). Dogs were sampled similarly (n = 36), except that all four paws were swabbed (n = 7 samples per dog taken at the same time that humans were sampled). The age and gender of humans surveyed in each family plus the number and breed of dogs in families with pets are summarized in *Table 1*. All samples were initially frozen at −20°C before they were transferred to the laboratory where they were stored at −80°C until they were subjected to DNA extraction, PCR of the variable region 2 (V2) of bacterial 16S rRNA genes and subsequent multiplex sequencing with an Illumina GAIIx instrument (Illumina, Inc., San Diego, CA; n = 969 samples used for the analyses reported, 74,855,127 total reads; average read length, 105 ± 19 nt). The resulting 16S rRNA dataset was analyzed using UniFrac (*Lozupone and Knight, 2005*), a phylogeny-based measure of the degree of similarity between microbial communities, to assess patterns of similarity within and between families across body sites.

### Strong effect of family membership on the human skin microbiota

Our results revealed that family unit had a strong effect on human microbial community composition across all body sites: at each site, family membership explained a large proportion of the variability in bacterial diversity as measured using Faith's phylogenetic diversity (PD) (*Faith, 1992*), suggesting that family members tend to harbor similar levels of bacterial diversity (*Table 2*). Composition was also significantly affected by family membership across all body sites such that communities were more similar within families than between them (*Table 3* and *Figure 1A–D*, dogs, if present, are shown together with family members). This pattern was strongest for skin, a body habitat in constant contact with the external environment. An analysis of similarity (ANOSIM) showed that R ranged from 0.21 to 0.62 for humans; the value was higher for dogs with R = 0.71 for forehead and R = 0.83 for paws. On the forehead and palms, all except the father-to-infant within-family distances were significantly smaller than between-family distances (*Figure 1A,B* and *Table 4*).

For tongue and feces, we observed effects at the level of family and partner, although generally of smaller magnitude than for skin sites (R < 0.30). The comparison between partners, which is nested within the comparison between family members, is the strongest for all body sites and likely drives similarities by family at these sites. In contrast, a weaker effect, or no effect, was observed for parent–offspring pairs (*Figure 1C,D*, R < 0.15). This effect seemed to depend primarily on the age of the child. Although parents may share significantly more similar tongue and gut communities with their own children than with other children at older ages (3–18 years), the same is not the case for parents and their infants. These findings agree with previous studies that found the fecal microbiota of teens to be more similar to that of their parents than to unrelated adults (*Yatsunenko et al., 2012*). However, our results also suggest that effects of cohabitation are insufficiently strong to overcome differences due to age (which are discussed in more detail in the section 'Effect of age on the human microbiota'), particularly between infants and adults, whose microbiota differ substantially (*Palmer et al., 2007*; *Koenig et al., 2011*).

We concluded that a shared environment may homogenize skin communities through contact with common surfaces (including each other). Likewise, it may be easier to exchange skin microbes

**Table 1.** Summary of the number, age classification, and gender of humans surveyed in each family and the number and types of animals in families with pets

| Adults (Sex) | Infants (Sex) | Adolescents (Sex) | Seniors (Sex) | Dogs (breed) | Other pets |
|---|---|---|---|---|---|
| 2 (M, F) | 1 (F) | 1 (F) | | | Cat |
| 2 (M, F) | | | | 1 (Unknown) | |
| 2 (M, F) | | | | | Cats |
| 2 (M, F) | | 1 (F) | | | |
| 2 (M, F) | | | | 2 (Unknown) | |
| 2 (M, F) | 1 (F) | | | 2 (Unknown) | |
| 2 (M, F) | 1 (F) | 1 (M) | | | |
| 2 (M, F) | | | | | Cats, guinea pigs |
| | | | 2 (M, F) | | |
| 2 (M, F) | | | | 2 (Unknown) | |
| 2 (M, F) | | | | 3 (Unknown) | |
| 2 (M, F) | | | | 1 (Unknown) | |
| 2 (M, F) | | | | | |
| 2 (M, F) | 1 (F) | | | | Cat, fish |
| 2 (M, F) | | 1 (M) | | | Cat |
| 2 (M, F) | | | | | |
| 2 (M, F) | | | | | |
| 2 (M, F) | | | | | Fish |
| 2 (M, F) | | 1 (F) | | 1 (Jack Russell Terrier) | Cat |
| 2 (M, F) | | | | 1 (Australian Cattle mix) | |
| 2 (M, F) | | 2 (M, F) | | | Cat, tarantula |
| 2 (M, F) | | | | 1 (Labrador/Golden mix) | Cat |
| 2 (M, F) | | | | 1 (Springer Spaniel) | Cat |
| 2 (M, F) | 1 (M) | 1 (M) | | | Cat |
| 2 (M, F) | | | | | Cats |
| 2 (M, F) | | | | | Reptiles, amphibians |
| 2 (M, F) | | | | | Cat |
| 2 (M, F) | | | | | Cats |
| 2 (M, F) | | | | | Fish |
| 2 (M, F) | 1 (F) | 2 (M, M) | | | |
| 2 (M, F) | | 2 (M, F) | | | Cat |
| 2 (M, F) | | 1 (F) | | | |
| 3 (M, F, M) | | 1 (M) | | | Cat |
| 2 (M, F) | | | | 1 (Border Collie) | |
| 2 (M, F) | | | | 2 (Kelpie, Standard Poodle) | |
| 2 (M, F) | 1 (F) | | | 1 (Border Collie mix) | |
| 2 (M, F) | | 1 (M) | | 2 (Boxer, Boxer) | |
| 2 (M, F) | | | | 2 (Labrador, Labrador) | |
| 2 (M, F) | | | | | |
| 2 (M, F) | | | | 2 (English Setter, Labrador) | |

*Table 1. Continued on next page*

*Table 1. Continued*

| Adults (Sex) | Infants (Sex) | Adolescents (Sex) | Seniors (Sex) | Dogs (breed) | Other pets |
|---|---|---|---|---|---|
| 2 (M, F) | | | | 2 (Labrador, Australian Shepherd/Spaniel mix) | Chickens |
| 2 (M, F) | | 2 (M, M) | | | Cats, rabbit, reptiles |
| 2 (M, F) | | 1 (F) | | | |
| 2 (M, F) | 1 (F) | | | 1 (German Shepherd mix) | |
| 2 (M, F) | | | | 1 (Bernese Mountain) | |
| 2 (M, F) | | 2 (M, F) | | | |
| 2 (M, F) | 1 (M) | 1 (M) | | | Cat |
| 2 (M, F) | | 3 (F, M, F) | | 1 (German Shepherd/ Malamute mix) | |
| 2 (M, F) | 1 (F) | 1 (F) | | | |
| 2 (M, F) | | | | 2 (Australian Shepherd, Australian Shepherd) | |
| 2 (M, F) | | 1 (M) | | 2 (Border Collie/German Shepherd mix, Labrador mix) | |
| 2 (M, F) | 1 (M) | | | | Cat |
| 1 (F) | | | 1 (M) | | Cat |
| 2 (M, F) | 1 (F) | | | 2 (Siberian Husky, Greater Swiss Mountain) | |
| | | | 2 (M, F) | | |
| | | | 2 (M, F) | | Cat |
| | | | 2 (M, F) | 1 (Unknown)* | |
| | | | 2 (M, F) | | |
| | | | 2 (M, F) | 1 (Unknown)* | |
| | | | 2 (M, F) | | |
| 106 | 12 | 26 | 15 | 36* | |

*These dogs were not sampled and thus not included in the total number of dogs.

Each row is one family and the last row contains the total for each column. Infants were considered to be individuals aged 0–12 months, children/adolescents as 1–17 years, adults as 18–59 years and seniors as ≥60 years

via exposure to home surfaces or indoor air (both of which are typically dominated by skin-associated microbes ; *Fierer et al., 2010*), than it is to exchange gut or mouth bacteria, potentially because skin surfaces may be less 'selective' environments compared to the gut or mouth environments.

## Effect of age on the human microbiota

Our results suggest that observed microbiota developmental dynamics depend on the body site under consideration (*Table 2* and *Figure 2*). Here, we define 'development' as the rate and pattern with which new hosts (i.e., infants and children) acquire adult-like microbiota over time. As noted previously (e.g., *Palmer et al., 2007*; *Koenig et al., 2011*; *Yatsunenko et al., 2012*), the development of the gut microbiota involves profound alterations in diversity and composition that take place over a relatively protracted timeframe (nominally, 0–3 years in age) (*Table 2* and *Figure 2*). Our study enables us to ask further whether similar dynamics are observed in the contemporaneously sampled oral and skin communities of the same individuals. For oral communities, diversity changed substantially with age (*Table 2*), with a notable increase between the age of 0 and 3 (*Figure 2B*), while compositional development, though significant, involved more subtle shifts than those observed over the same age range in the gut (*Table 5*). On the skin, diversity and composition (i.e., here, strictly membership) changed relatively little with age (*Table 2* and *Figure 2*). Interestingly, however, using a distance metric that emphasizes abundance (weighted UniFrac) reveals a strong developmental

shift in skin microbiota on the forehead (*Figure 2A*), a trend driven in part by changes in the relative abundance of dominant taxa rather than the acquisition/loss of unique taxa with age. This result is consistent with earlier studies (*Somerville, 1969*; *Leyden et al., 1975*). For example, we see the relative abundance of Propionibacteria significantly increase on the forehead with age (*Table 6*), which has been shown to be associated with increasing levels of sebum production (*Leyden et al., 1975*; *McGinley et al., 1980*). The lack of a significant effect of age with unweighted UniFrac and the small amount of the variance in microbial diversity explained by age suggests developmental dynamics affected more by environmental exposures (ostensibly, to adult skin microbiota) than by age-associated shifts in the selective landscape (e.g., via introduction of solid food, emergence of teeth, etc.).

## Taxa shared by cohabiting partners

We next examined which groups of taxa are shared more between cohabiting partners than by adults from different families. *Figure 1E* shows an example of the specific taxa that are shared within and between adult partners on the right palm. The taxa driving these differences on the palm are lineages commonly reported in surveys of the human skin microbiota such as Propionibacteria (*Costello et al., 2009*; *Grice et al., 2009*). Two of these taxa, Prevotella and Veillonella, are primarily associated with the human oral community (*Nasidze et al., 2009*), suggesting that at least for cohabiting couples, oral-skin transfer may be moderately frequent.

**Table 2.** Summary of the optimal linear mixed model explaining microbial phylogenetic diversity of body sites in relation to the main factors

| Age group | Body site | Term | Type | Estimate | SE | %Variability |
|---|---|---|---|---|---|---|
| All | Palms (L&R) | NoDog | Fixed | −0.19 | 0.06 | |
| | | Family | Random | | | 38.98 |
| | | Age | Random | | | 17.14 |
| | | Plate | Random | | | 4.34 |
| | | FamSize | Random | | | 0.38 |
| | | Lane | Random | | | 0 |
| | | BS | Random | | | 0 |
| | | Residual | | | | 39.16 |
| | Forehead | Family | Random | | | 24.85 |
| | | Age | Random | | | 15.41 |
| | | Lane | Random | | | 8.50 |
| | | FamSize | Random | | | 7.81 |
| | | Plate | Random | | | 6.33 |
| | | Residual | | | | 37.10 |
| | Fecal | Age | Random | | | 45.64 |
| | | Plate | Random | | | 7.15 |
| | | FamSize | Random | | | 3.08 |
| | | Family | Random | | | 2.24 |
| | | Lane | Random | | | $2.97 \times 10^{-9}$ |
| | | Residual | | | | 41.88 |
| | Oral | Age | Random | | | 35.92 |
| | | Family | Random | | | 14.54 |
| | | Lane | Random | | | 3.64 |
| | | Plate | Random | | | $4.60 \times 10^{-10}$ |
| | | FamSize | Random | | | 0 |
| | | Residual | | | | 45.90 |

*Table 2. Continued on next page*

*Table 2. Continued*

| Age group | Body site | Term | Type | Estimate | SE | %Variability |
|---|---|---|---|---|---|---|
| Adults | Palms (L&R) | NoDogs | Fixed | −0.22 | 0.068 | |
| | | Male | Fixed | −0.11 | 0.031 | |
| | | Family | Random | | | 46.90 |
| | | Plate | Random | | | 2.77 |
| | | FamSiz | Random | | | $5.17 \times 10^{-11}$ |
| | | Lane | Random | | | $2.16 \times 10^{-12}$ |
| | | BS | Random | | | 0 |
| | | Residual | | | | 50.33 |
| | Forehead | NoDogs | Fixed | −0.27 | 0.087 | |
| | | Family | Random | | | 34.91 |
| | | FamSize | Random | | | 9.08 |
| | | Plate | Random | | | 0.80 |
| | | Lane | Random | | | 0 |
| | | Residual | | | | 55.21 |
| | Fecal | Family | Random | | | 20.20 |
| | | Lane | Random | | | 9.77 |
| | | FamSize | Random | | | 0 |
| | | Plate | Random | | | 0 |
| | | Residual | | | | 70.03 |
| | Oral | Family | Random | | | 26.16 |
| | | FamSize | Random | | | 14.59 |
| | | Lane | Random | | | 3.43 |
| | | Plate | Random | | | 3.42 |
| | | Residual | | | | 52.40 |

The model takes into account the variability between age groups (Age), families (Family), family sizes (FamSize), sequencing lanes (Lane), and primer plates (Plate). Variability between left and right palms (BS) is also controlled for in the palm model. The table gives parameter estimates and standard errors for the significant terms in the model and the percentage of explained variability for each of the random effects ordered from highest to lowest.

Because these taxa are also found in the gut, we tried to determine the level at which the palm communities contained taxa derived from either oral or fecal sources. Using SourceTracker (*Knights et al., 2011*), we estimated that on average, ~11% of the palm community is likely from oral sources, as opposed to <2% from fecal sources. Given that oral bacteria can persist on skin for at least 8 hr (*Costello et al., 2009*), we do not know whether these patterns are due to repeated inoculation from oral-skin contact or a true establishment of oral microbes on skin habitats. However, these results do suggest that close physical contact (such as that between cohabiting couples) can affect the taxonomic composition of the skin and may explain why these communities are more similar.

## Effect of cohabitation with dogs on the human skin microbiota

Interestingly, the similarity in the microbiota of cohabiting individuals extends beyond human-to-human relationships to pet-to-pet and even human-to-pet relationships. The patterns of similarity between cohabiting dogs mimic that of cohabiting people, with skin (fur) sites showing the greatest degree of similarity (*Figure 1A–D*). Moreover, from a microbial perspective, the skin communities of adults are on average more similar to those of their own dog(s) than to other dogs (*Figure 3*). Thus, we further explored the effect of dogs on the overall bacterial diversity and composition of their cohabiting owners in more detail. A principal components analysis (PCoA) of the human skin communities

**Table 3.** Summary of a permutational analysis of variance (PERMANOVA) assessing the effect of family membership on unweighted UniFrac distances between families

| Body site | Source of variation | Df | MS | Pseudo-F | P | $\sqrt{V}$ |
|---|---|---|---|---|---|---|
| Palms (L&R) | Family membership | 69 | 0.3274 | 2.0702 | **0.001** | 0.21 |
| | Residual | 210 | 0.15815 | | | 0.40 |
| Forehead | Family membership | 70 | 0.23758 | 1.4603 | **0.001** | 0.18 |
| | Residual | 91 | 0.1627 | | | 0.40 |
| Fecal | Family membership | 66 | 0.212 | 1.2759 | **0.001** | 0.14 |
| | Residual | 92 | 0.16615 | | | 0.41 |
| Oral | Family membership | 68 | 0.12544 | 1.6803 | **0.001** | 0.15 |
| | Residual | 96 | 0.07465 | | | 0.27 |

$\sqrt{V}$ are the estimates of the components of variance for the factors. Statistically significant effects are in bold.

did not show strong clustering of the sites by dog-owning status in the three main axes, suggesting that dogs do not have a large effect. However, once age was accounted for, we found that dog ownership also affects the skin communities of adults, such that dog owners share more similar communities than expected by chance (*Table 5*). This effect is not seen when the weighted UniFrac measure is used, suggesting that dog-owners share similar communities mainly due to the addition of rare rather than abundant taxa. Such effects were detected in adults, but not infants or seniors, and may be due to behavioral differences between age groups that were not measured, such as variation in levels of contact with dogs. Alternatively, the presence of strong age affects as indicated earlier combined with small sample sizes of children with and without dogs may have obscured our ability to detect a significant effect of dogs.

Because dogs appear to have the largest effect on the skin communities of their cohabiting adult owners, we then explored the differences in the number of shared phylotypes (OTUs), and the overall bacterial diversity on the skin of adults with dogs as well as adults with children. Adults who have dogs share more bacterial phylotypes with each other than they do with adults who do not have dogs (*Figure 4*, top right). Having a dog, then, has an effect of similar size on the number of taxa shared in human skin communities as the effect of living together (i.e., two people who have dogs but do not live together share, on average, about as many phylotypes as two people who live together but who do not have a dog). Adults who have a dog and live together share the greatest number of skin phylotypes while adults who neither have a dog nor live together share the least. We tested the effects of gender, pet ownership, and cohabitation of children using a linear mixed effects model, taking into account the variability in diversity due to family membership, age, and technical differences (e.g., sequencing lane). Of these factors, only dog ownership and gender significantly affected diversity (*Table 7*). Adults who own dogs tended to have a higher diversity of bacteria on their skin (hands and forehead) than those without dogs (*Figure 4*, top left, p<0.001, Student's t-test with 10,000 Monte Carlo simulations; *Table 2*). It should also be noted that consistent with previous studies (*Fierer et al., 2008*), we found that adult females tend to have a higher diversity of bacteria on their hands than adult males (*Table 2*).

## Effects on other body habitat microbiota and of other environmental microbial reservoirs

In contrast to the skin communities, effects of gender or dogs were not detected in the gut or oral communities (*Table 7*). In fact, none of the tested factors were identified as important in these communities. Curiously, owning other types of indoor pets (i.e., cats [which were not sampled for this study]) did not have a significant effect on the diversity (*Table 7*), overall similarity (*Table 5*), or amount of taxa shared between the skin communities of adult partners (p=0.92, Wilcoxon test). Although family size (i.e., having a child or children) did seem to have a significant effect on whether individuals shared more similar skin communities, having a child in our study cohort also did not have a comparable effect to age or dog ownership on community similarity or diversity

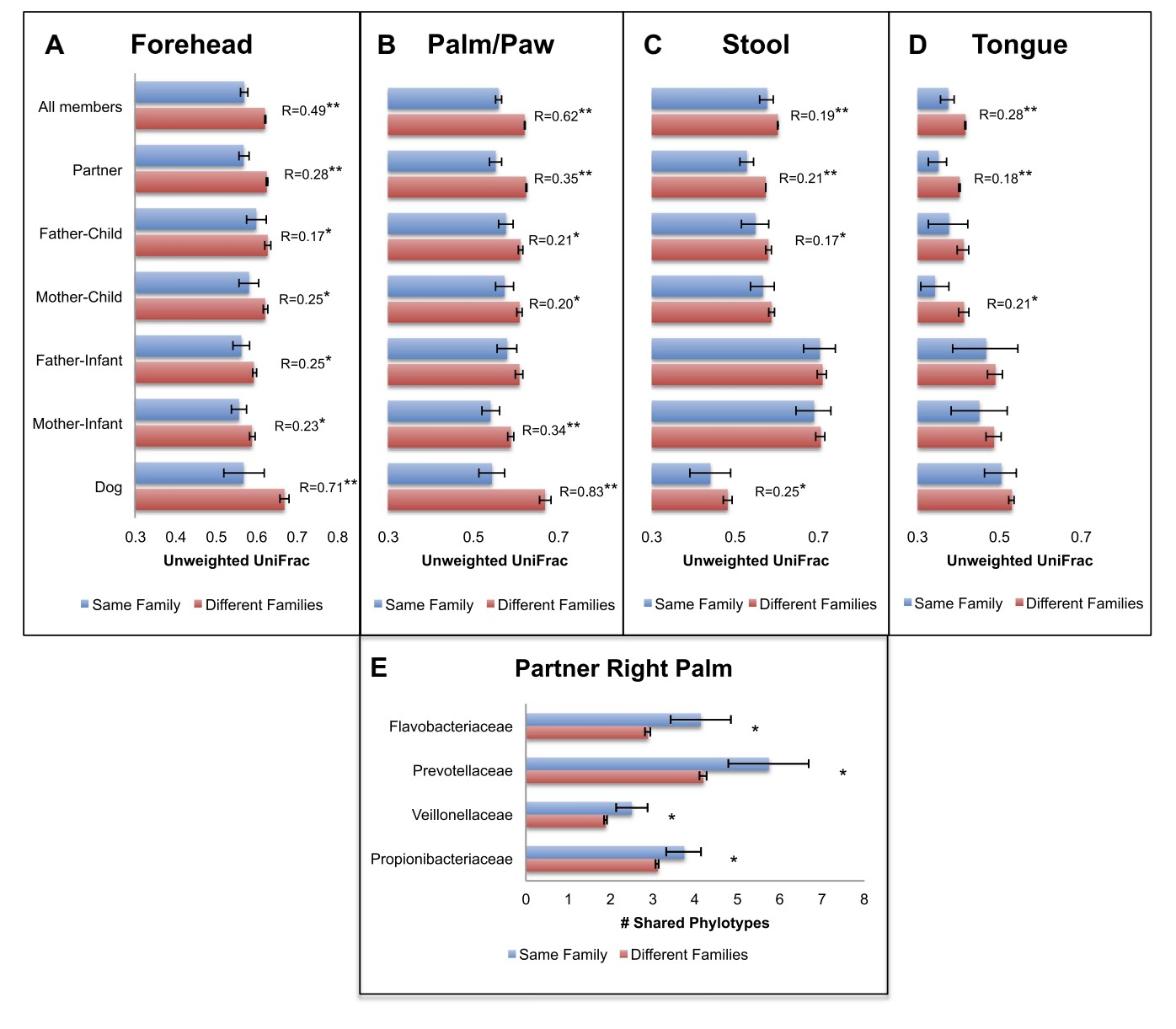

**Figure 1**. Community similarity within and between families across body sites, and taxa contributing to these differences. Panels (**A**–**D**) show average unweighted UniFrac distances between family members (blue) and between members of different families (red). 'Child' refers to all offspring aged 3–18 years who cohabit with the parents. 'Infants' were considered to be individuals aged 0–12 months. Palm/Paw refers to the right palm in the human comparisons and the back left paw in the dog comparison. Although there are distinguishable differences between the left and right palm communities within and across individuals (**Fierer et al., 2008**), the same analysis using the left palms showed a similar pattern (**Table 2**) and neither composition nor diversity were different enough between palms or among the four dog paws to affect the overall patterns. Mean ± 95% CI and R values (ANOSIM) are shown. *p<0.05 and **p<0.001 based on 10,000 permutations. Panel (**E**) shows the families of bacteria that exhibit the greatest differences in the number of phylotypes (OTUs) shared within and between adult partners on the right palm. Bars represent the average number of shared phylotypes for a given bacterial family within partners from the same family (blue) and between partners of different families (red). Mean ± 95% CI shown. *p<0.05 after Bonferroni correction (Wilcoxon test).

(**Figure 4**, bottom, p=0.05 for diversity; **Tables 5 and 7**). For example, family size only explained a small proportion of the variability in diversity across all body sites (oral: 0%, palms: <1%, fecal: 3%, forehead: 8%) compared to age (oral: 36%, palms: 17%, fecal: 46%, forehead: 15%). Although the mean difference in the number of phylotypes shared between couples paired from different

**Table 4.** Summary of ANOSIM analyses of the differences between within-family and between-family community comparisons using unweighted (unwtd) and weighted (wtd) UniFrac

| Comparison | Body site | Fam (N) | Ind (N) | R (unwtd) | p | R (wtd) | p |
|---|---|---|---|---|---|---|---|
| Families | Forehead | 60 | 151 | 0.49 | *<0.0001* | 0.010 | 0.39 |
| | Right palm | 60 | 141 | 0.62 | *<0.0001* | 0.18 | **0.0008** |
| | Left palm | 56 | 122 | 0.61 | *<0.0001* | 0.30 | *<0.0001* |
| | Fecal | 59 | 151 | 0.19 | *<0.0001* | 0.068 | 0.056 |
| | Oral | 59 | 155 | 0.28 | *<0.0001* | 0.32 | *<0.0001* |
| Spouses | Forehead | 60 | 114 | 0.28 | *<0.0001* | 0.091 | **0.0048** |
| | Right palm | 59 | 105 | 0.35 | *<0.0001* | 0.11 | **0.0042** |
| | Left palm | 55 | 91 | 0.31 | *<0.0001* | 0.13 | **0.002** |
| | Fecal | 59 | 114 | 0.21 | *<0.0001* | 0.077 | **0.0032** |
| | Oral | 59 | 117 | 0.18 | *<0.0001* | 0.18 | *<0.0001* |
| Father–Child | Forehead | 14 | 31 | 0.17 | **0.025** | 0.026 | 0.36 |
| | Right palm | 14 | 31 | 0.21 | **0.0085** | 0.11 | 0.096 |
| | Fecal | 14 | 33 | 0.17 | 0.053 | 0.066 | 0.20 |
| | Oral | 14 | 33 | 0.07 | 0.069 | 0.23 | **0.0051** |
| Mother–Child | Forehead | 14 | 31 | 0.25 | **0.0024** | 0.018 | 0.41 |
| | Right palm | 14 | 30 | 0.20 | **0.013** | 0.0019 | 0.51 |
| | Fecal | 14 | 33 | 0.10 | 0.11 | 0.10 | 0.095 |
| | Oral | 14 | 33 | 0.21 | **0.0047** | 0.26 | **0.0012** |
| Father-Infant | Forehead | 12 | 24 | 0.25 | **0.0034** | 0.030 | 0.99 |
| | Right Palm | 12 | 21 | 0.16 | **0.016** | 0.0063 | 0.96 |
| | Fecal | 12 | 23 | 0.057 | 0.96 | −0.10 | 0.97 |
| | Oral | 12 | 24 | 0.061 | 0.61 | −0.0086 | 0.70 |
| Mother–Infant | Forehead | 12 | 23 | 0.23 | **0.0025** | −0.025 | 0.99 |
| | Right palm | 12 | 21 | 0.34 | *<0.0001* | 0.012 | 0.87 |
| | Fecal | 12 | 23 | 0.066 | 0.94 | −0.036 | 0.86 |
| | Oral | 12 | 24 | 0.092 | 0.23 | 0.074 | 0.42 |
| Dogs | Forehead | 12 | 22 | 0.71 | *0.0002* | 0.56 | **<0.0001** |
| | Back left paw | 12 | 23 | 0.83 | *<0.0001* | 0.68 | **<0.0001** |
| | Fecal | 12 | 25 | 0.25 | **0.0079** | 0.20 | **0.039** |
| | Oral | 12 | 25 | 0.25 | **0.017** | 0.30 | **0.023** |

'Child' refers to all offspring aged 3–18 years who cohabit with the parents. 'Infant' is considered to be an individual aged 0–12 months. The number of families and individuals used in each analysis is shown. Statistically significant comparisons (p<0.05) are bolded. Those both statistically significant and of relatively large magnitude (R>0.25) are bolded and italicized.

families with and without children was significant, the size of the difference was minor (<2 phylotypes) and is likely due to the large number of observations in the within and between categories (2757 and 3071 respectively), which gave us the power to detect very small effect sizes (d = 0.085 based on a power analysis given these sample sizes).

## Mechanistic considerations

One possible explanation for the large effect of dogs in comparison to children and other pets may be that individuals with dogs harbor taxa different from those without dogs, largely due to the presence of dog-derived bacterial taxa on their skin and presumably from frequent direct contact. One of the main taxa driving the pattern of similarity between dog owners is a family of Betaproteobacteria (Methylophilaceae), a group that was also highly abundant (4.6%) in the

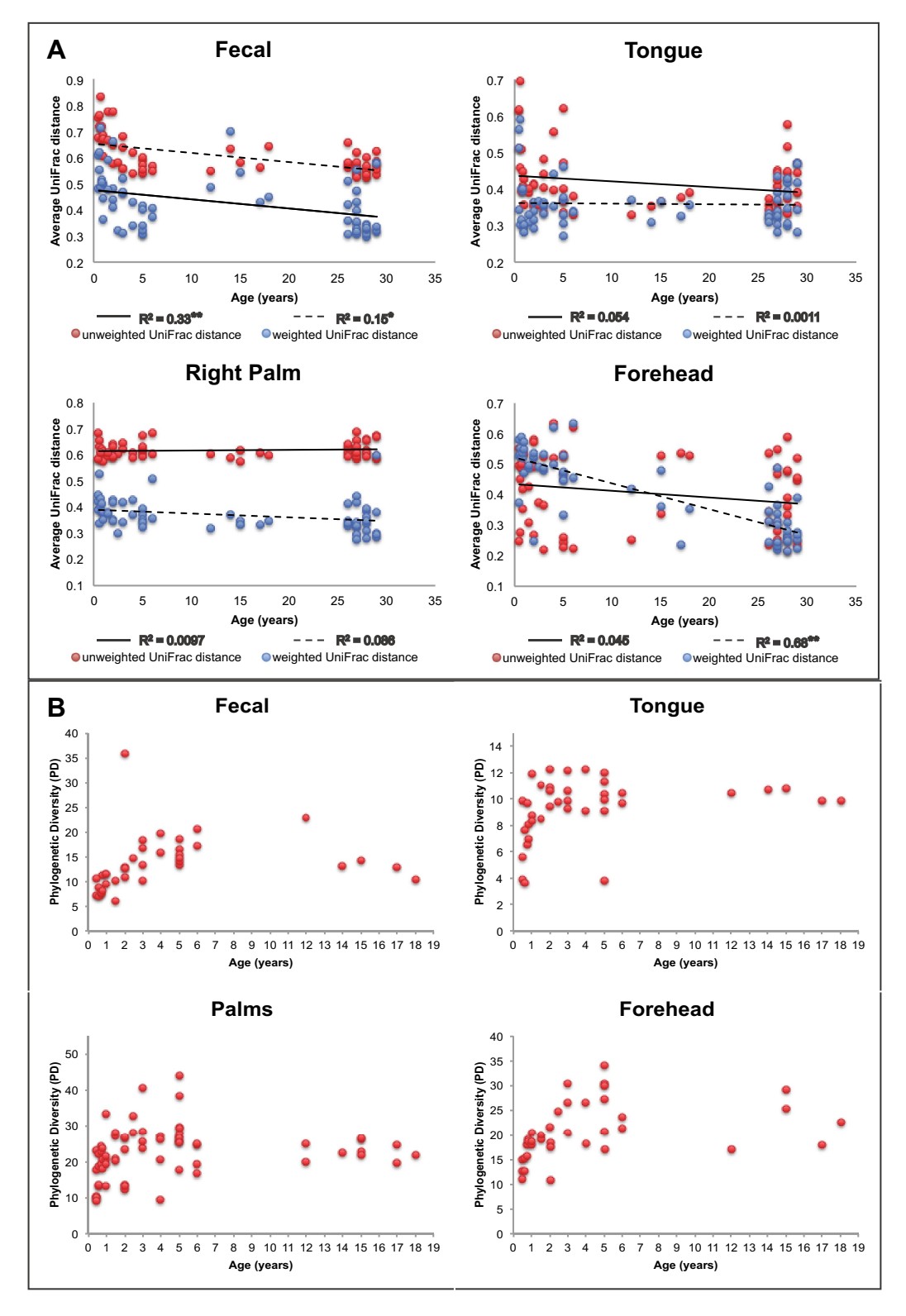

**Figure 2**. Approach towards or departure from the 'adult' state in each body site with age. (**A**) Each point represents the average distance (unweighted UniFrac in red; weighted UniFrac in blue) between each participant and all other participants in the 'adult' age bracket. Here we define baseline 'adult' as 30–45 years in age (the results are not sensitive to this threshold). $R^2$ values (linear regression model) are shown. *p<0.01, **p<0.001. (**B**) Phylogenetic diversity (PD) of the communities on each body site is plotted for all of the offspring in the study (aged 0–18 years).

**Table 5.** Summary of a permutational distance-based linear model testing the effect of age (Age), gender (Sex), pet ownership (Dog or Cat), and family size (FamSize) on unweighted and weighted UniFrac distance (measures of community dissimilarity)

| | AgeGroup | Body site | Best model | AICc | Psuedo-F | p | r² |
|---|---|---|---|---|---|---|---|
| Unweighted | All | Palms (L&R) | **(+)Age** | −453.49 | 5.9182 | **0.001** | **2.2** |
| | | | (+)Dog | −455.94 | 4.4837 | 0.001 | 1.19 |
| | | | (+)FamSize | −454.95 | 2.0378 | 0.001 | 0.62 |
| | | Forehead | **(+)Age** | −264.31 | 2.7579 | **0.001** | **1.69** |
| | | Fecal | **(+)Age** | −269.34 | 4.3775 | **0.001** | **2.71** |
| | | Oral | **(+)Age** | −386.58 | 2.4774 | **0.002** | **1.5** |
| | Adults | Palms (L&R) | **(+)Dog** | −284.03 | 4.303 | **0.001** | **2.43** |
| | | Forehead | **(+)Dog** | −166.1 | 2.3878 | **0.001** | **2.33** |
| | | Fecal | None | NA | NA | NA | NA |
| | | Oral | None | NA | NA | NA | NA |
| | Infants | Palms (L&R) | None | NA | NA | NA | NA |
| | | Forehead | None | NA | NA | NA | NA |
| | | Fecal | None | NA | NA | NA | NA |
| | | Oral | None | NA | NA | NA | NA |
| | Seniors | Palms (L&R) | None | NA | NA | NA | NA |
| | | Forehead | None | NA | NA | NA | NA |
| | | Fecal | None | NA | NA | NA | NA |
| | | Oral | None | NA | NA | NA | NA |
| Weighted | All | Palms (L&R) | **(+)Age** | −727.66 | 23.597 | **0.001** | **9.05** |
| | | | (+)Sex | −730.84 | 5.218 | 0.001 | 1.57 |
| | | | (+)FamSize | −733.39 | 4.5795 | 0.002 | 0.39 |
| | | Forehead | **(+)Age** | −418.2 | 19.463 | **0.001** | **15.11** |
| | | | (+)Sex | −423.78 | 7.6939 | 0.001 | 1.29 |
| | | | (+)FamSize | −424.89 | 3.1675 | 0.028 | 0.22 |
| | | Fecal | (+)Age | −353.08 | 2.4044 | 0.061 | 1.51 |
| | | Oral | **(+)FamSize** | −416.32 | 3.4073 | **0.019** | **2.05** |
| | Adults | Palms (L&R) | **(+)Sex** | −462.24 | 4.5058 | **0.002** | **5.36** |
| | | | (+)FamSize | −464.11 | 3.9149 | 0.003 | 1.09 |
| | | | (+)Dog | −465.65 | 3.5858 | 0.003 | 0.22 |
| | | Forehead | **(+)Sex** | −295.94 | 4.3628 | **0.008** | **4.18** |
| | | Fecal | None | NA | NA | NA | NA |
| | | Oral | None | NA | NA | NA | NA |

This analysis was performed for the entire data set (All) as well as separately for age groups. The terms from the best model for each body site are shown, along with the percent of total variation explained (r²). Those terms of statistically significant and largest effect are bolded for each body site.

mouths of the dogs in this study, consistent with a common occurrence of oral–skin transfer between dogs and their owners. Other taxa include several families of Actinobacteria and a family of Acidobacteria commonly associated with soil (*Lauber et al., 2009*), all of which were present on the paws and forehead of dogs, although in relatively low abundance (<1%). In addition, characterization of the dog oral and 'skin' (fur and paws) microbiota revealed a greater diversity of taxa than described in humans (*Table 8*). Whereas human skin tends to be dominated by a few taxa at relatively high abundance (namely Propionibacteriaceae, Streptococcaceae and Staphylococcaceae), dog paws and forehead harbor a more even mixture of taxa commonly found

**Table 6.** Summary of taxon abundances (%) present on the forehead for each age group

| Taxon | Infants | Children/Adolescents | Adults | Seniors |
|---|---|---|---|---|
| *Actinobacteria* | | | | |
| *Propionibacteriaceae** (8.78 × 10⁻¹³) | **6.3** | **12** | **51** | **31** |
| *Corynebacteriaceae* | 0.7 | 2.0 | 4.2 | 9.3 |
| *Micrococcaceae** (0.0072) | **2.0** | **2.7** | **1.2** | **2.5** |
| *Bacteroidetes* | | | | |
| *Prevotellaceae** (7.02 × 10⁻⁹) | **7.7** | **5.6** | **1.4** | **1.5** |
| *Porphyromonadaceae** (8.53 × 10⁻¹²) | **2.8** | **3.1** | **0.4** | **0.8** |
| *Flavobacteriaceae* | 1.2 | 1.4 | 1.1 | 2.0 |
| *Bacteroidaceae* | 0.5 | 1.4 | 1.1 | 1.9 |
| *Firmicutes* | | | | |
| *Streptococcaceae** (5.32 × 10⁻³⁷) | **47** | **27** | **5.5** | **8.2** |
| *Staphylococcaceae** (0.0088) | **2.1** | **2.6** | **12** | **4.3** |
| *Carnobacteriaceae** (5.99 × 10⁻²⁴) | **4.8** | **3.4** | **0.5** | **0.7** |
| *Veillonellaceae** (1.25 × 10⁻¹³) | **4.6** | **1.9** | **0.7** | **1.3** |
| *Alphaproteobacteria* | | | | |
| *Sphingomonadaceae* | 0.9 | 1.4 | 1.3 | 0.9 |
| *Betaproteobacteria* | | | | |
| *Neisseriaceae* | 2.4 | 4.1 | 2.3 | 6.9 |
| *Comamonadaceae* | 0.4 | 1.2 | 1.5 | 2.9 |
| *Gammaproteobacteria* | | | | |
| *Pasteurellaceae** (1.64 × 10⁻⁶) | **5.3** | **7.0** | **1.3** | **2.4** |
| *Moraxellaceae* | 0.7 | 2.2 | 1.7 | 1.8 |

*A significant effect of age (p<0.05 after Bonferroni correction; exact p-values are shown in parentheses). Infants were considered to be individuals aged 0–12 months, children/adolescents as 1–17 years, adults as 18–59 years and seniors as ≥60 years.
Family level abundances of >1% were subjected to ANOVA analysis in QIIME.

in a variety of host-associated environments including mammalian gut (Enterobacteriaceae, Fusobacteriaceae), mouth (Porphyromonadaceae, Veillonellaceae), and skin (Propionibacteriaceae, Staphylococcaceae), as well as free-living environments such as soil and water (e.g., Hyphomicrobiaceae and Sphingomonadaceae) (*Tables 6, 8, and 9*). This evenness and diversity of taxa found on dog skin may reflect frequent exposure of these sites to many different sources of microbes, or behavioral differences. The dog gut and tongue communities, on the other hand, harbor microbial communities that are somewhat similar in diversity and composition yet distinct from those in the human counterparts (*Figure 5*). Collectively, these data suggest that our pets not only harbor a diverse microbial community, but also shed a diverse set of microbiota that may in turn influence our own microbial composition.

## Prospectus

Given that recent studies in gnotobiotic and other animal models show pervasive effects of the microbiota on metabolism, immunity, and other aspects of our biology, it is intriguing to consider that who we cohabit with, including companion animals, may alter our physiological properties by influencing the consortia of microbial symbionts that we harbor in and on our various body habitats, and in particular, our skin habitats. One example relates to the hygiene hypothesis, which posits that a broad range of microbial exposures helps educate our developing immune systems to tolerate a variety of environmental antigens, thereby reducing risk for atopic disorders such as asthma and food allergies. Recent studies link early exposure to pets to decreased prevalence of allergies, respiratory conditions, and other immune disorders in later stages of development

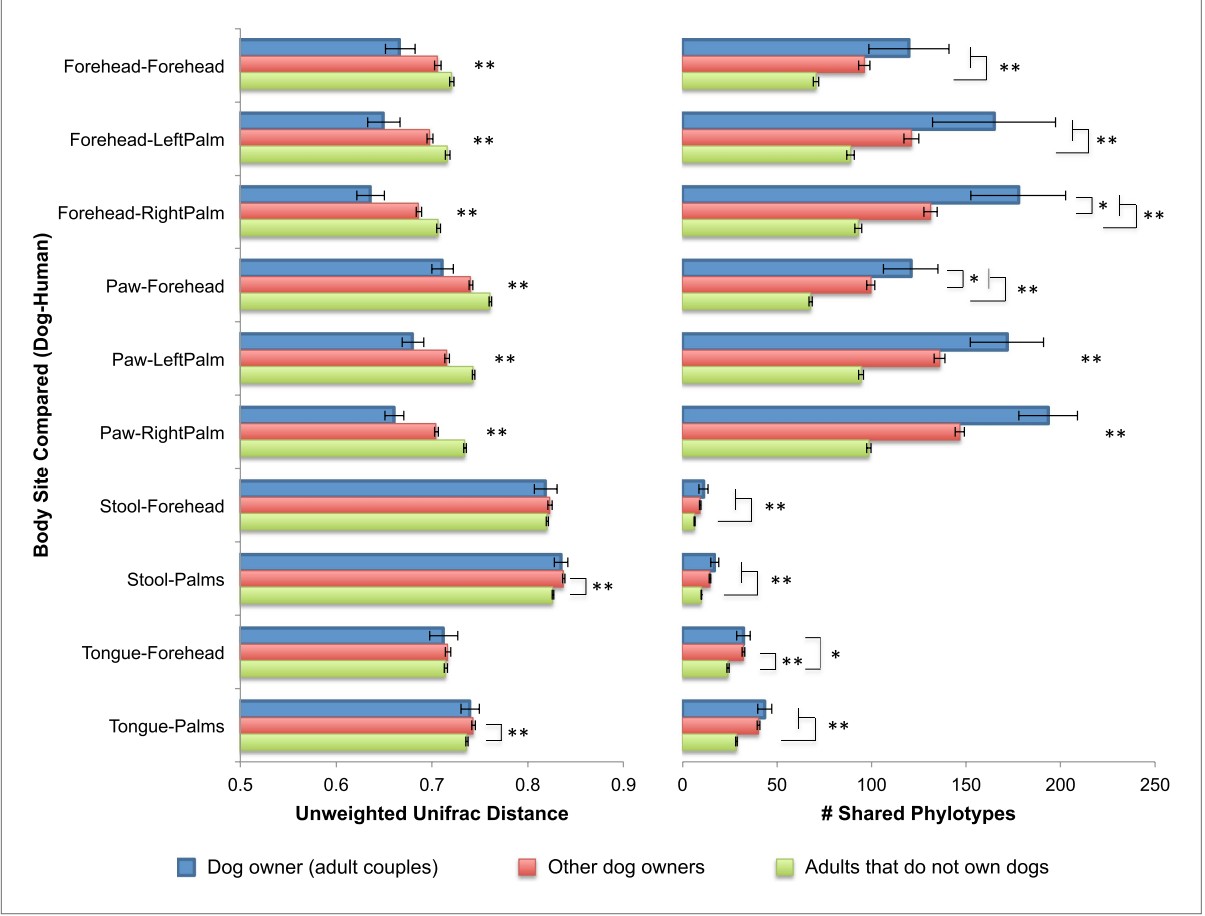

**Figure 3**. Community similarity and phylotype sharing between dogs-owners and their dogs. The left panel shows the average unweighted UniFrac distance between adult dog-owners and their dogs (blue), between dog-owners and other (not their own) dogs (red), and between adults who do not own dogs and dogs (green). The right panel shows the number of phylotypes shared for the same categories. Comparisons are labeled on the y-axis such that the first body site listed corresponds to the dog and the second site corresponds to the human. Mean ± 95% CI shown. The presence of asterisks lacking brackets indicates that all pairwise comparisons within that group are significant. Generally, dog-owners tend to share more similar communities and more phylotypes with their own dogs than with other dogs. *p<0.05, **p<0.001 after Bonferroni correction (Wilcoxon test).

(*Havstad et al., 2011*) and skin microbes in particular are now receiving more focus as important players in immune regulation (*Naik et al., 2012*). Given the potential of skin as a collector and integrator of shared environmental bacteria as demonstrated in this study, identifying exactly how such communities can be functionally affected by environmental exposures may help us better understand how they may be deliberately manipulated in order to prevent or treat disease. Epidemiologic studies of the impact of environmental factors on physiological variations and disease predispositions would be enhanced by integrating microbiological surveys, including time series studies during the first years of postnatal life. These efforts would be timely as we seek to understand the impact of Westernization on human biology and to delineate, from an anthropologic perspective, how different cultural traditions and lifestyles relate to our microbial ecology (*Benezra et al., 2012*).

## Materials and methods

### DNA extraction and multiplex sequencing
Each sample was processed using methods and procedures described in previous publications (*Hamady et al., 2008*; *Caporaso et al., 2011*). DNA was extracted from each swab using the

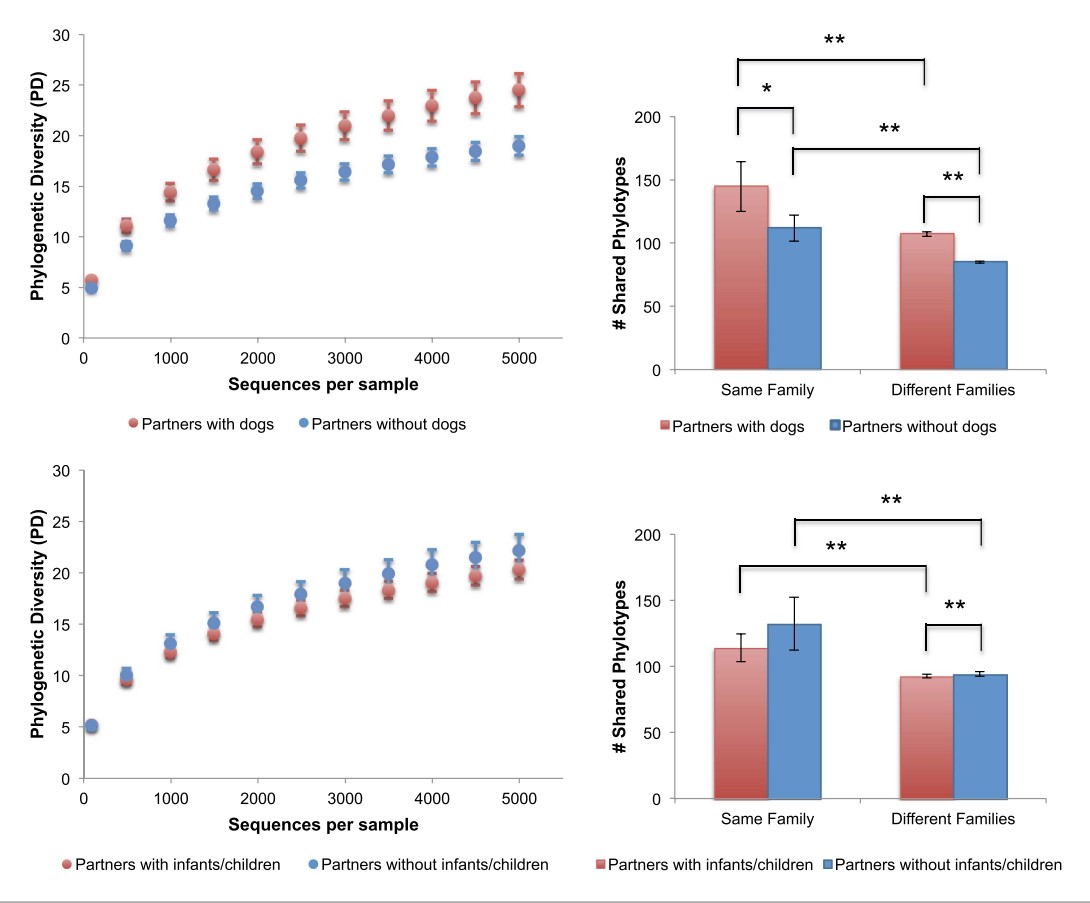

**Figure 4**. Alpha diversity and shared phylotypes in couples with and without dogs and children. The left panels show rarefaction curves for skin communities of couples (including seniors) who have dogs (top, in red), those without dogs (top, in blue), couples (excluding seniors) with infants/children (bottom, in red), and those without infants/children (bottom, in blue). Mean ± 95% CI shown. The right panels show the average number of phylotypes shared among individuals from the same categories shown in the left panels. Mean ± 95% CI shown. *p<0.05, **p<0.001 after Bonferroni correction (Wilcoxon test).

MOBIO PowerSoil DNA isolation kit (MO BIO Laboratories, Inc., Carlsbad, CA) according to manufacturer instructions with modifications. For each sample, the V2 region of bacterial 16S rRNA genes was amplified in triplicate reactions using the primers F27 (5′-AGAGTTTGATCCTGGCTCAG-3′) and R338 (5′-TGCTGCCTCCCGTAGGAG T-3′) barcoded with a unique 12-base error-correcting Golay code for multiplexing. PCR reactions contained 13 µl MO BIO PCR water, 10 µl 5 Prime Hot Master Mix, 0.5 µl each of the forward and reverse primers (10 µM final concentration), and 1.0 µl genomic DNA. Reactions were held at 94°C for 3 min to denature the DNA, run for 35 cycles of amplification at 94°C for 45 s, 50°C for 60 s, and then 72°C for 90 s, and completed with a final extension step of 10 min at 72°C. Amplicons were processed using the MO BIO Ultra Clean-htp 96-well PCR clean up kit and quantified using Picogreen dsDNA reagent in 10-mM Tris buffer (pH 8.0). Equal amounts of amplicons from each reaction for a given sample were pooled, followed by gel purification and ethanol precipitation. Multiplex DNA sequencing was performed with a Illumina GAIIx instrument located in the Center for Genome Sciences and Systems Biology at Washington University School of Medicine. The resulting DNA sequences, OTU table, and associated sample metadata have been deposited in the QIIME database (http://www.microbio.me/qiime/) under the study ID 979 (Song_2012_family_study).

## Sequence processing and analysis

Sequence data were processed with QIIME v1.4.0-dev (*Caporaso et al., 2010*) as previously described (*Yatsunenko et al., 2012*). Sequences were demultiplexed and quality filtered using

default QIIME parameters, and 16S rRNA operational taxonomic units (OTUs) were picked using a closed reference OTU picking procedure (QIIME script pick_reference_otus_through_otu_table. py). Briefly, sequences were clustered against the Greengenes database (reference collection, 2011 release) (http://greengenes.lbl.gov/) at 97% identity and those failing to match within this threshold were discarded. Taxonomy was assigned to the retained clusters (OTUs) based on the Greengenes reference sequence and the Greengenes tree was used for all downstream phylogenetic community comparisons. For the 1076 samples, the number of sequences per sample ranged from 1 to 300,473, with a mean of 54,475 sequences per sample (total: 58,615,414). Such large variability in the number of sequences per sample is typical for studies employing high-throughput sequencing methods, the causes of which have not yet been systematically tested to our knowledge. To standardize sequence counts across samples, samples with <5000 sequences per sample were removed. Remaining samples were rarefied to 5000 sequences and further filtered by eliminating samples that had a high probability of being mislabeled (e.g., labeled skin, but likely a tongue or fecal sample; detected using the script supervised_learning.py). The remaining 969 samples were used for all downstream analyses. Phylogenetic diversity (PD) was computed and rarefaction analyses were conducted using the QIIME scripts multiple_rarefaction.py, alpha_diversity.py and collate_ alpha.py. Because there are no generally accepted methods for 'denoising' Illumina sequence data, alpha diversity estimates such as PD and OTU counts may be overestimated due to sequencing error. However, overestimation should not affect relative differences in diversity. Analyses of community similarity (β-diversity) were performed by calculating pairwise distances using the phylogenetic metric UniFrac (*Lozupone and Knight, 2005*). The resulting distance matrices were used for principle coordinates analyses (PCoA).

**Table 7.** Summary of a linear mixed effects analysis on the response of bacterial diversity across the body sites using the full data set and then filtered to include just adults

| Age group | Body site | Fixed effects | Random effects | Model | ΔAIC | Pr(Chi) |
|---|---|---|---|---|---|---|
| All | Palms (L&R) | Dog + Cat + Sex + Kid | BS + Ag + Fa + FS + Pl + La | Full | 0 | NA |
| | | | | (−)Dog | 2.91 | **0.027** |
| | | | | (−)Cat | −5.60 | 1 |
| | | | | (−)Sex | 1.73 | 0.053 |
| | | | | (−)Kid | −5.62 | 1 |
| | Forehead | Dog + Cat + Sex + Kid | Ag + Fa + FS + Pl + La | Full | 0 | NA |
| | | | | (−)Dog | 0.57 | 0.11 |
| | | | | (−)Cat | −5.25 | 1 |
| | | | | (−)Sex | 0.97 | 0.085 |
| | | | | (−)Kid | −4.22 | 1 |
| | Fecal | Dog + Cat + Sex + Kid | Ag + Fa + FS + Pl + La | Full | 0 | NA |
| | | | | (−)Dog | −0.049 | 0.16 |
| | | | | (−)Cat | −0.15 | 0.17 |
| | | | | (−)Sex | 0.33 | 0.13 |
| | | | | (−)Kid | −0.13 | 0.17 |
| | Oral | Dog + Cat + Sex + Kid | Ag + Fa + FS + Pl + La | Full | 0 | NA |
| | | | | (−)Dog | 1.22 | 0.072 |
| | | | | (−)Cat | −1.95 | 0.82 |
| | | | | (−)Sex | −1.88 | 0.73 |
| | | | | (−)Kid | −1.78 | 0.64 |

*Table 7. Continued on next page*

*Table 7. Continued*

| Age group | Body site | Fixed effects | Random effects | Model | ∆AIC | Pr(Chi) |
|---|---|---|---|---|---|---|
| Adults | Palms (L&R) | Dog + Cat + Sex + Kid | BS + Fa + FS + Pl + La | Full | 0 | NA |
| | | | | (−)Dog | 4.09 | **0.014** |
| | | | | (−)Cat | −5.29 | 1 |
| | | | | (−)Sex | 4.11 | **0.013** |
| | | | | (−)Kid | −5.69 | 1 |
| | Forehead | Dog + Cat + Sex + Kid | Fa + FS + Pl + La | Full | 0 | NA |
| | | | | (−)Dog | 2.52 | **0.033** |
| | | | | (−)Cat | −4.99 | 1 |
| | | | | (−)Sex | −0.37 | 0.20 |
| | | | | (−)Kid | −4.38 | 1 |
| | Fecal | Dog + Cat + Sex + Kid | Fa + FS + Pl + La | Full | 0 | NA |
| | | | | (−)Dog | 0.30 | 0.13 |
| | | | | (−)Cat | 0.24 | 0.13 |
| | | | | (−)Sex | −1.44 | 0.45 |
| | | | | (−)Kid | −0.42 | 0.21 |
| | Oral | Dog + Cat + Sex + Kid | Fa + FS + Pl + La | Full | 0 | NA |
| | | | | (−)Dog | −0.29 | 0.19 |
| | | | | (−)Cat | −6.50 | 1 |
| | | | | (−)Sex | −7.22 | 1 |
| | | | | (−)Kid | −5.41 | 1 |

The factors tested are co-habitation of dogs (Dog), cats (Cat), children (Kid) and host gender (Sex). The base model takes into account the variability between age groups (Ag), families (Fa), family sizes (FS), sequencing lanes (La), and primer plates (Pl). Variability between left and right palms (BS) is also controlled for in the palm model. The change in model fit resulting from exclusion of each fixed effect based on Akaike's Information Criterion (AIC) is shown. Statistically significant values are in bold text.

## Statistical analyses

### Identification of factors of main effect

We used a permutational multivariate analysis of variance (PERMANOVA) using the PERMANOVA+ add on (*Anderson et al. 2008*) to Plymouth Routines In Multivariate Ecological Research (PRIMER v6) package (*Clarke and Gorley, 2006*). The PERMANOVA analysis was based on the unweighted UniFrac dissimilarity matrix, type III partial sums of squares, and 999 random permutations of the residuals under the reduced model to determine whether communities differ significantly between families. We then used a custom script for an analysis similar to an analysis of similarities (ANOSIM) in R version 2.13.2 (*R Development Core Team, 2011*) to test the hypothesis that within-family communities are more similar than between-family communities. Using the unweighted UniFrac distance matrix, distances were grouped as 'within family' or 'between family' according to criteria appropriate for the comparison (e.g., for partners, the within family group consisted of distances between male and female adult individuals in the same household and the between family group consisted of distances between all combinations of those same adult males and females from different families). Significance levels were calculated by comparing the R statistic against the distribution generated from 10,000 permutations of the randomized dataset.

To determine whether age, gender, pet ownership, and family size (i.e., cohabitation of children) explain differences in community composition, we used a distance-based linear model analysis with 999 permutations using the package DistLM in the PERMANOVA+ add-on to PRIMER. Models were

**Table 8.** Summary of taxon abundances (%) on the external body sites of dogs

| Taxon | Back left paw | Back right paw | Front left paw | Paws averaged | Forehead |
|---|---|---|---|---|---|
| *Actinobacteria* | | | | | |
| *Corynebacteriaceae* | 1.8 | 1.3 | 0.7 | **1.2** | **1.0** |
| *Microbacteriaceae* | 4.0 | 4.2 | 4.3 | **4.2** | **1.8** |
| *Micrococcaceae* | 2.2 | 2.6 | 2.4 | **2.4** | **1.7** |
| *Nocardioidaceae* | 3.3 | 3.4 | 3.2 | **3.3** | **1.5** |
| *Propionibacteriaceae* | 3.0 | 3.9 | 3.1 | **3.3** | **4.5** |
| *Bacteroidetes* | | | | | |
| *Bacteroidaceae* | 2.1 | 1.9 | 2.6 | **2.2** | **3.2** |
| *Porphyromonadaceae* | 2.8 | 1.6 | 1.7 | **2.0** | **5.7** |
| *Prevotellaceae* | 1.1 | 1.8 | 3.0 | **2.0** | **2.4** |
| *Flavobacteriaceae* | 2.6 | 2.5 | 2.5 | **2.5** | **3.3** |
| *Flexibacteraceae* | 1.6 | 1.7 | 1.7 | **1.7** | **1.5** |
| *Firmicutes* | | | | | |
| *Staphylococcaceae* | 1.4 | 2.0 | 1.0 | **1.5** | **1.2** |
| *Streptococcaceae* | 1.2 | 1.6 | 1.5 | **1.4** | **2.8** |
| *Lachnospiraceae* | 1.1 | 0.9 | 1.7 | **1.3** | **1.2** |
| *Veillonellaceae* | 0.3 | 0.5 | 1.8 | **0.9** | **0.7** |
| *Fusobacteria* | | | | | |
| *Fusobacteriaceae* | 2.3 | 1.7 | 1.8 | **2.0** | **3.8** |
| *Alphaproteobacteria* | | | | | |
| *Bradyrhizobiaceae* | 0.6 | 0.8 | 0.5 | **0.6** | **1.0** |
| *Hyphomicrobiaceae* | 1.1 | 1.2 | 1.1 | **1.2** | **0.7** |
| *Methylobacteriaceae* | 0.4 | 0.7 | 0.3 | **0.5** | **1.0** |
| *Sphingomonadaceae* | 5.3 | 6.6 | 5.2 | **5.7** | **6.1** |
| *Betaproteobacteria* | | | | | |
| *Comamonadaceae* | 1.9 | 1.9 | 1.5 | **1.8** | **2.6** |
| *Oxalobacteraceae* | 1.5 | 1.8 | 1.5 | **1.6** | **1.3** |
| *Neisseriaceae* | 1.3 | 1.0 | 1.4 | **1.2** | **2.8** |
| *Gammaproteobacteria* | | | | | |
| *Enterobacteriaceae* | 2.2 | 5.1 | 5.9 | **4.4** | **2.5** |
| *Oceanospirillaceae* | 2.3 | 1.0 | 3.1 | **2.1** | **0.3** |
| *Pasteurellaceae* | 2.4 | 2.5 | 2.3 | **2.4** | **6.9** |
| *Moraxellaceae* | 1.9 | 1.4 | 1.6 | **1.6** | **2.2** |
| *Pseudomonadaceae* | 7.1 | 4.8 | 5.5 | **5.8** | **3.4** |

Family level abundances >1% are shown. The front right paw was sampled but failed to amplify and is therefore not shown.

developed using a step-wise process of adding and removing the factors. Model selection was based on Akaike's Information Criterion with a second order bias correction (AICc).

## Description of main effects

We examined the effect of pet ownership, gender, and the cohabitation of children on the bacterial diversity (measured as phylogenetic diversity [PD]) of each body site using a linear mixed effects model including age group, family membership, family size, sequencing lane, and primer plate as random factors. Variability between left and right palms was also controlled for in the palm model. For each body site, we began with the full model including all random and fixed factors, and fixed factors were

**Table 9.** Summary of taxon abundances (%) present on the palms for each age group

| | Infants | Children/ Adolescents | Adults | Seniors |
|---|---|---|---|---|
| *Actinobacteria* | | | | |
| Corynebacteriaceae | 0.7 (0.6) | 3 (2.3) | 4.2 (3.9) | 4.3 (4.0) |
| Micrococcaceae | 4.7 (3.9) | 3.9 (3.9) | 2.8 (2.6) | 2.7 (2.8) |
| Propionibacteriaceae* (0.00016) | **3.1 (2.2)** | **11 (11)** | **27 (27)** | **20 (15.9)** |
| *Bacteroidetes* | | | | |
| Bacteroidaceae | 0.5 (0.6) | 1.6 (1.4) | 1.6 (3.3) | 3.4 (8.7) |
| Flavobacteriaceae | 1 (1.4) | 1.9 (1.4) | 2.7 (2.0) | 3.8 (4.1) |
| Porphyromonadaceae | 2 (2.1) | 1.9 (1.5) | 1.4 (0.7) | 0.6 (1.0) |
| Prevotellaceae | 5.6 (4.2) | 4.2 (5.7) | 3.1 (3.0) | 2.1 (1.7) |
| *Firmicutes* | | | | |
| Carnobacteriaceae* ($1.15 \times 10^{-11}$) | **6.4 (5.8)** | **5 (3.5)** | **1.7 (1.9)** | **1 (0.7)** |
| ClostridialesFamilyXI.IncertaeSedis | 0.2 (NA) | 1.5 (NA) | 1 (NA) | 1.4 (NA) |
| Lachnospiraceae* ($8.57 \times 10^{-5}$) | **0.3 (0.5)** | **0.9 (1.3)** | **1 (1.5)** | **3.1 (7.2)** |
| Lactobacillaceae | 0.1 (NA) | 0.2 (NA) | 1.5 (NA) | 4.2 (NA) |
| (Ruminococcaceae*) ($1.17 \times 10^{-5}$) | **NA (0.2 )** | **NA (0.7)** | **NA (0.8)** | **NA (3.5)** |
| Staphylococcaceae | 3.2 (2.1) | 5.1 (7.2) | 6.7 (7.3) | 2.8 (2.2) |
| Streptococcaceae* ($3.91 \times 10^{-10}$) | **49 (54)** | **27 (26)** | **15 (16)** | **13 (9.1)** |
| Veillonellaceae* ($1.76 \times 10^{-7}$) | **5.5 (4.5)** | **2.2 (2.1)** | **1.7 (2.0)** | **1.9 (2.1)** |
| *Fusobacteria* | | | | |
| Fusobacteriaceae | 1.6 (1.6) | 1.8 (1.2) | 1.4 (1.2) | 1 (1.0) |
| *Betaproteobacteria* | | | | |
| Comamonadaceae* ($9.90 \times 10^{-5}$) | **0.7 (0.3)** | **0.7 (1.0)** | **1.5 (1.7)** | **3.6 (3.7)** |
| Neisseriaceae | 2.6 (2.4) | 3.5 (2.3) | 1.6 (1.1) | 2 (1.6) |
| *Gammaproteobacteria* | | | | |
| Moraxellaceae | 1.5 (0.7) | 1.2 (1.7) | 3.2 (2.7) | 3.3 (2.7) |
| Pasteurellaceae* (0.017) | **2.6 (3.3)** | **4.4 (2.9)** | **1.8 (1.6)** | **1.2 (1.1)** |
| Pseudomonadaceae* (0.049) | **0.2 (0.2)** | **0.6 (0.7)** | **1.1 (1.0)** | **2.4 (3.0)** |
| (Enterobacteriaceae*) (0.027) | **NA (0.6)** | **NA (0.8)** | **NA (0.8)** | **NA (2.7)** |

*A significant effect of age (p<0.05 after Bonferroni correction; exact p-values are shown in parentheses). Shown only for the right palm (left palm showed similar trends). Infants were considered to be individuals aged 0–12 months, children/adolescents as 1–17 years, adults as 18–59 years and seniors as ≥60 years. Abundances for the left palm are shown in parentheses. Family level abundances of greater than 1% were subjected to ANOVA analysis in QIIME. Taxa present at >1% on the left palm but <1% on the right are shown in parentheses.

subsequently removed in a step-wise manner using the function 'drop1' in R. Akaike's Information Criterion (AIC) values and a chi-square test were used to select the best model for each body site. An increase in the AIC value indicates that the removed factor significantly worsened the fit of the model. All modeling was performed using the function 'lmer' in the R-package 'lme4' (*Bates et al., 2008*). Differences in alpha diversity between groups were subsequently tested using a t-test with Monte Carlo simulations on the dataset rarefied to 5000 sequences (compare_alpha_diversity.py in QIIME v1.5.0-dev).

To describe changes in the microbial community with age, distances were calculated between each participant and all participants within specified age groups (the core groups of adults were considered 30–45 years old and elderly participants ≥65 years old), averaged for each participant, and then plotted against their age using the QIIME script categorized_dist_scatterplot.py. A linear regression model was fitted to the distance plots using R. Analyses of differences in the number of shared phylotypes between groups were performed using the Wilcoxon test in R.

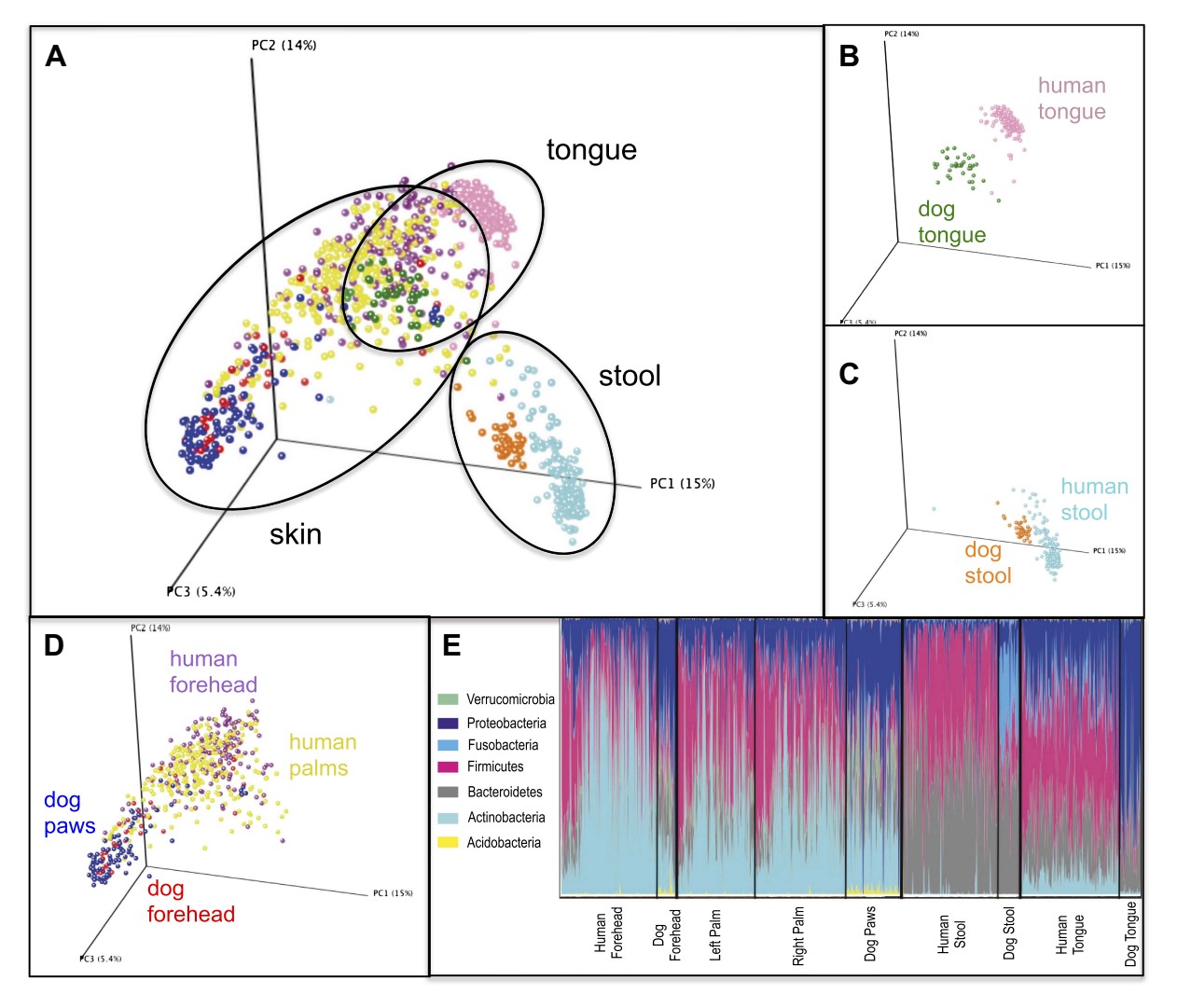

**Figure 5**. Variation within and between the communities of skin, oral, and fecal samples from humans and dogs. Panel (**A**) shows a PCoA plot of all the body habitats, using unweighted UniFrac distances of human and dog samples, rarefied at 5000 sequences/sample. Panels (**B**–**D**) show select body habitats from the full plot. Panel (**E**) shows a summary of the taxa shaded by relative abundance at the phylum level broken down by specific body habitat; the seven most abundant taxa are shown in the legend.

For all body sites, we tested for differences in taxon abundances across the age groups using an analysis of variance (ANOVA) (otu_category_significance.py in QIIME). Infants were considered to be individuals aged 0–12 months, children/adolescents as 1–17 years, adults as 18–59 years and seniors as ≥60 years. Most of the subjects in the child/adolescent category were between the ages of 1 and 6 years, none were between 7 and 11 years, and five participants were between the ages of 12 and 17 years. Due to the low sample size in the latter age range, post-pubescent subjects/teens were not split into a separate category. Exclusion of these five subjects from the analysis did not significantly affect the results. In all appropriate analyses, p values were adjusted for the number of comparisons made using the Bonferroni method.

## Acknowledgements
We thank S Whitehead for help with participant recruitment and sampling. This work was supported in part by the Crohns and Colitis Foundation of America, the National Institutes of Health, and the Howard Hughes Medical Institute.

## Additional information

### Funding

| Funder | Grant reference number | Author |
|---|---|---|
| Howard Hughes Medical Institute | | Rob Knight |
| Crohn's and Colitis Foundation of America | | Jeffrey I Gordon |
| National Institutes of Health | HG4872, HG4866 | Rob Knight |

The funders had no role in study design, data collection and interpretation, or the decision to submit the work for publication.

### Author contributions

SS, Acquisition of data, Analysis and interpretation of data, Drafting or revising the article; CL, RK, Conception and design, Analysis and interpretation of data, Drafting or revising the article; EKC, CAL, JIG, NF, Conception and design, Drafting or revising the article; GH, DBL, Acquisition of data, Drafting or revising the article; JGC, DK, JCC, Analysis and interpretation of data, Drafting or revising the article; SN, Conception and design, Acquisition of data, Drafting or revising the article

### Ethics

Human subjects: Participants were recruited and sampled using methods approved by the Institutional Review Board at the University of Colorado (Protocol#: 0409.13). Informed consent and consent to publish was obtained from all subjects. The Helsinki guidelines were followed.
Animal experimentation: All animals were handled and sampled according to protocol #1102.06, approved by the approved institutional animal care and use committee (IACUC) of the University of Colorado.

## Additional files

### Major datasets

The following dataset was generated:

| Author(s) | Year | Dataset title | Dataset ID and/or URL | Database, license, and accessibility information |
|---|---|---|---|---|
| Song S, Lauber C, Costello EK, Lozupone CA, Humphrey G, Berg-Lyons D, Caporaso JG, Knights D, Clemente JC, Nakielny S, Gordon JI, Fierer N, Knight R | 2012 | Data from: Cohabiting family members share microbiota with 1 one another and with their dogs. | Study ID: 797 (Song_2012_family_study) | Available in the QIIME database at http://www.microbio.me/qiime/. Standard used to collect data: The data collected in this study follows the MIMARKS guidelines for reporting 16S survey data. |

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
