## [Decision Letter]

Thank you for choosing to send your work entitled “Cohabiting family members share microbiota with one another and with their dogs” for consideration at *eLife*. Your article has been evaluated by a Senior editor, Detlef Weigel, who also acted as Reviewing editor, and two outside reviewers.

The Reviewing editor and the other reviewers discussed their comments before we reached this decision, and the Reviewing editor has assembled the following comments to help you prepare a revised submission.

Song and colleagues present a fascinating, albeit descriptive, 16S rDNA-based survey of human co-habitation, spanning 60 families with children, dogs, both, or neither. As expected, they appear to find that individuals living together have increased similarity in their associated microbial communities, with a markedly stronger effect on skin microbiota, compared to oral or gut microbiotae. An important advance over earlier work is that dog owners may “swap” microbes with their pets, again most clearly shown for skin microbiota. There are, however, three major concerns:

1. The first one relates to the statistical methods used. Rather than pairwise comparisons, the authors need to apply a general linear mixed model that includes as many covariates as possible, such as collection dates, family size, geographic origin, urban/rural sites, presence of other pets, sequencing runs, and so on.

2. The second major concern is that although the authors suggest potential reasons for the apparent sharing of microbiota between dogs and their owners (see “Mechanistic considerations”), they do not explicitly test the hypothesis that the transfer of rare taxa drives these trends. For example, the authors might filter out taxa that are more commonly found in the dog microbiota, or found at much lower abundance in humans, and then test if that significantly changes the observed trends. Another approach might be to use weighted metrics instead (e.g., Figure 1), as unweighted data can be greatly influenced by rare but novel taxa.

3. Third, based on the hypothesis that “more microbes are shared between individuals who share a greater number of potential microbial sources”, one expects to see an analysis based on the number of human individuals in the household, since other humans are sources, too. This has been relevant for the hygiene hypothesis (more children in a family corresponds with lower incidence of hay fever in later children). Thus, did the number of people in the household explain any of the variance in these analyses? And do later-birth order children host more diverse microbiotae? Also, what can you say about children with dogs versus children without dogs (i.e., not just adults or household members) – are there positive effects of dog ownership?

Some extra analysis is also required for the following points:

A. It is stated that skin microbiota does not change with age as much as stool microbiota, based on a comparison of individuals of different ages. Later on it is stated that skin microbiota tends to reflect environmental microbiota, in agreement with the hypothesis that humans pick up microbes from their dogs. So one interpretation of the absence of a strong age effect is that the skin is more dominated by the environment than the stool microbiota.

B. Are the taxa shared by dog owning adults actually found on dogs? Are the exact same taxa picked up (at the OTU level)? Also, were there effects of the other pets or were the sample sizes simply too small?

C. Finally, Bruce Levin and colleagues asked similar questions based on *E. coli* strains isolated from household members and their pets in the 1980s – it would be a shame to omit a comparison to these earlier findings. Please take a look at PMID:6376625, for instance.

---

## [Author Response]

*1) The first one relates to the statistical methods used. Rather than pairwise comparisons, the authors need to apply a general linear mixed model that includes as many covariates as possible, such as collection dates, family size, geographic origin, urban/rural sites, presence of other pets, sequencing runs, and so on*.

We thank the reviewers for this important suggestion. We have addressed this point as follows:

We have applied a linear mixed model and we have added the results of this, as well as additional analyses, to our paper. We are happy to report that these analyses further support all of our main conclusions. A brief summary of the results is included below.

Note: Because none of our response variables were expected to follow a particular alternative distribution, such as a binomial or poisson distribution, we used a linear mixed model instead of a generalized linear mixed model. Response variables that were not normally distributed approached normality after transformation and the transformed values were used in the linear models.

A permutational multivariate analysis of variance (PERMANOVA) identified family membership as having a significant effect on community similarity (unweighted UniFrac distance), further supporting our conclusion that communities are more similar between family members than across families.

Age, sex, pet ownership, and cohabitation of children were modeled to explain community similarity using a permutational distance-based linear model approach. This analysis identified age as an important factor for all of the body sites. Once age was controlled for (i.e., the analysis was run on the adult age group separately), dog ownership was the only factor that explained community similarity for the forehead and palms, while none of the factors explained fecal or oral community similarity.

To determine the effect of pet ownership, sex, and child cohabitation on the bacterial diversity of body sites, we used a linear mixed model with family membership, age group, family size, sequencing lane, and primer plate as random factors. Additional factors were not included due to missing data. We expected differences between sequencing lanes and plates because sample collection and processing were performed sequentially, such that blocks of families, as well as samples from our elderly subjects, were run on the same lane and plate, and we now know that family membership and age both affect patterns of community similarity. This analysis supported our conclusion that dog ownership has an effect on the diversity of skin (forehead and palms), such that not having dogs is associated with lower diversity. Consistent with previous studies, sex was also found to have an effect such that females tend to have a higher bacterial diversity on their palms than males.

*2) The second major concern is that although the authors suggest potential reasons for the apparent sharing of microbiota between dogs and their owners (see “Mechanistic considerations”), they do not explicitly test the hypothesis that the transfer of rare taxa drives these trends. For example, the authors might filter out taxa that are more commonly found in the dog microbiota, or found at much lower abundance in humans, and then test if that significantly changes the observed trends. Another approach might be to use weighted metrics instead (e.g., Figure 1), as unweighted data can be greatly influenced by rare but novel taxa*.

We have added an analysis of the effect of dog ownership on the weighted UniFrac measure of dissimilarity, the results of which are now appended to Table 5. Briefly, while the effect of dogs on the unweighted UniFrac distance between participants is significant, it is not for the weighted measure, which indicates that rare rather than abundant taxa are driving these patterns.

*3) Third, based on the hypothesis that “more microbes are shared between individuals who share a greater number of potential microbial sources”, one expects to see an analysis based on the number of human individuals in the household, since other humans are sources, too. This has been relevant for the hygiene hypothesis (more children in a family corresponds with lower incidence of hay fever in later children). Thus, did the number of people in the household explain any of the variance in these analyses? And do later-birth order children host more diverse microbiotae? Also, what* can *you say about children with dogs versus children without dogs (i.e., not just adults or household members) – are there positive effects of dog ownership*?

Our linear mixed model showed that family size only explains a very small proportion of the variability in PD across the body sites (Table 2). A distance-based linear model also testing the effect of family size has now been added to the analyses, the results of which are shown in Table 5.

In response to the question of whether older children host more diverse microbiota, a figure showing scatterplots of PD against age has been added (Figure 3).

The PD of children with dogs was compared with that of children without dogs. Because infants and very young children have been shown to have very different communities from older children, children under the age of 3 years were excluded from this analysis. An effect of owning dogs was not detected, but this is likely due to the low sample size. Of the 20 children aged 3–18 years, only three had dogs and two of these three were from the same family. Therefore these results are not included in the manuscript.

Some extra analysis is also required for the following points:

*A) It is stated that skin microbiota does not change with age as much as stool microbiota, based on a comparison of individuals of different ages. Later on it is stated that skin microbiota tends to reflect environmental microbiota, in agreement with the hypothesis that humans pick up microbes from their dogs. So one interpretation of the absence of a strong age effect is that the skin is more dominated by the environment than the stool microbiota*.

We have added this observation to the new specific section on age effects.

*B) Are the taxa shared by dog owning adults actually found on dogs? Are the exact same taxa picked up (at the OTU level)? Also, were there effects of the other pets or were the sample sizes simply too small*?

Yes, taxa shared by dog-owners were also found on dogs, as we had already noted. There were several cat owners, but cats were not sampled. Therefore, we could not determine whether there were taxa that were shared among cat-owners. However, cats did not have a significant effect on PD or community similarity among cat-owners. The sample size was small for other pets.

*C) Finally, Bruce Levin and colleagues asked similar questions based on E. coli strains isolated from household members and their pets in the 1980s – it would be a shame to omit a comparison to these earlier findings. Please take a look at PMID:6376625, for instance*.

We thank the reviewer for pointing out this pioneering work in which fecal E. coli isolates were compared among parents, children, dogs and cats (28 individual hosts) composing five families (6). It found that isolate types were more likely to be shared within than between families. It also describes one family in which two isolate types were common to both human and non-human family members. We now reference these earlier findings.